# The Influence of a 9-Week Movement Program on the Body Composition of 7- to 8-Year-Old Schoolchildren in the Eastern Cape of South Africa

**DOI:** 10.3390/ijerph20031762

**Published:** 2023-01-18

**Authors:** Mere Idamokoro, Anita E. Pienaar, Barry Gerber, Maria M. van Gent

**Affiliations:** 1Focus Area of Physical Activity, Sport and Recreation (PhASRec), Faculty of Health Sciences, North-West University, Potchefstroom 2531, South Africa; 2Department of Human Movement Sciences, University of Fort Hare, Alice 5700, South Africa

**Keywords:** BMI, anthropometric characteristics, early childhood, fundamental motor skills, movement program, obesity, overweight, rural school children, sustainably

## Abstract

Pediatric obesity has become a growing global epidemic which has negative health consequences, including for South African children. This study aimed to determine the immediate and sustainable influences of a 9-week movement program on the body composition of 7 to 8-year-old school children in a rural area of South Africa. A two group, pre-test, post-test and re-test after six months experimental design was used to compare anthropometric measurements of the intervention group (IG) and control group (CG). Ninety-three schoolchildren (IG = 57; CG = 36) participated in the study. A 9-week movement program was followed twice a week for 30 min during school hours with an emphasis on improving BMI. Hierarchical Linear Modelling (HLM) was used to analyze the data with time, sex and group as predictors. Effect sizes was computed based on the Cohen’s d to assess the practical significance of findings. The intervention positively changed the waist circumference. The subscapular skinfold and BMI showed statistical and practically significant sustainable changes because of the intervention, although gender influenced these effects. School based movement interventions, focusing on improving fundamental movement skills (FMS), have the potential to contribute to a healthier BMI, skinfold thickness and circumferences among young children.

## 1. Introduction

Pediatric obesity is one of the most serious public health challenges of the 21st century, affecting every country globally [1]. In the last 40 years, obesity among school-age children and youths has risen more than 10-fold, from 11 million to 124 million (2016 estimates) globally [1]. This epidemic specifically increased rapidly in low- and middle-income countries, especially in Northern- and Southern Africa countries, the Middle East and the Pacific Islands [2]. The Global Action Plan focuses on combating this obesity and overweight burden and all governments are urged to meet the international targets set out to combat this epidemic [2]. These international targets address the underlying roots of obesity which includes its definition, causes and prevention strategies [2].

Pediatric obesity contributes to a range of health challenges that can be physical, social, academic and psychological [3,4]. Serious health risk factors are furthermore reported to be associated with overweight and obesity such as gastrointestinal disorders, cardiovascular disease (CVD), joint and muscular disorders, respiratory problems, type 2 diabetes (T2D), psychological issues, hypertension and mortality risks [5,6,7].

The South African Primary Schools’ Anthropometric Survey and The Health of the Nation Study [8] revealed an estimated increase from 1.2% to 13% in overweight and in obesity between 1994–2002 among South African children and adolescence, with the occurrence still rising. The South African National Health and Nutrition Examination Survey (SANHANES-1) [9] furthermore reported a prevalence of 13.5% overweight and obesity in children aged 6–14 years in 2013, which is higher than the 10% global prevalence in children. In addition, the more recent HAKSA 2016, 2019 (*Healthy Active Kids South Africa*, HAKSA, 2016, 2019) report cards and the Child Gauge, again report rising prevalence’s among South African children and adolescents [10,11,12]. This prevalence of overweight and obesity in children and adolescents from South African, which is a developing country, is equivalent with that of developed nations and among the highest in Africa [8,13].

More recent studies conducted in different Provinces of South Africa confirmed increases in overweight and obesity in primary school children which are associated with negative health challenges. Significant increases (*p* < 0.05) in overweight and obesity among school children aged 6–19 years in the Eastern Cape Province of South Africa are reported [14], where an overall 14.8% (21.1% girls and 7.8% boys) were overweight and 2.8% (4.6% girls and 1.0% boys) obese. A study in the Western Cape Province of South Africa reported that obesity is highly prevalent in school learners aged 7–18 years, with 22.9% being overweight/obese [15]. This study furthermore revealed that being female or black African increased the odds of being overweight/obesity and these odds were also associated with adverse cardio-metabolic risk profiles in school-going children [15]. In the North West Province, a longitudinal study of 6–9 year old school children reported an overall increase from 12.5% at baseline to 16.7% prevalence of overweight/obesity over a 3-years period [16]. In the North-West Province of South Africa (SA), a recent study by de Waal and Pienaar [17] also revealed a prevalence of 9.97% among children of persistent overweight/obesity over a longitudinal period of 7 school years. This was also associated with poorer and deteriorating weight bearing motor skills and balancing skills in such children, which again can negatively influence healthy behaviors. Strategies, such as improving physical activity (PA) levels and nutrition, are therefore urgently needed to combat this epidemic among South African children [10,11].

A systematic review including 50 studies reviewed childhood obesity prevention programs to determine the effectiveness of interventions that were intended to prevent obesity in children, as assessed by changes in Body Mass Index (BMI) [18]. The content of the interventions was based on educational, health promotion, psychological/family/behavioral therapy/counselling/management which focus on diet, physical activity, or lifestyle support. The immediate outcomes of the interventions were promising, but lack of sustainability effects was evident. It was recommended that interventions should be embedded into the operating systems and ongoing practices of children, rather than implementing interventions that are resource intensive which cannot be maintained over a long-term.

Schools in this regard, are reported to be a suitable place for interventions that can promote PA in children, particularly in disadvantaged communities [19]. To increase children’s PA participation in schools and to contribute to the prevention of overweight/obesity successfully, it is necessary to increase the active time in physical education (PE) classes where these programs are embedded [20]. Additionally, studies conducted in this respect have shown that school-based interventions can be effective in preventing overweight/obesity and promoting PA among school children [20]. Researchers [21] also expatiated on the benefits of school-based physical activity intervention to lessen the increase of body fat in normal-weight and in overweight/obese children. In this regard, the school-based intervention of 10-weeks that were conducted by these researchers, involving 898 children (458 boys, 440 girls) aged 8–11 years from disadvantaged communities in South Africa, revealed a positive effect on the children’s body composition [21].

It is, however, clear that prevention should start at a young age [13]. It is suggested [18] that a promising strategy to combat overweight and obesity is to include healthy eating, PA and healthy body composition, increased sessions for PA and development of fundamental motor skills (FMS) throughout the school week. Competency in FMS has been proposed to interact with perceptions of motor competence and health related fitness, resulting in improved levels of PA and subsequently combatting obesity from childhood to adulthood [22]. Poor motor competency at a young age is furthermore associated with behavior that is conducive to obesity [23,24,25,26,27] and strongly influences healthy behavior. Therefore, a better understanding of this association to the obesity healthcare challenge is still needed [17]. Young children are still at an optimal age for motor skill development, thus preventing a decrease in motor coordination over time that may add to health challenges [17]. The National report on the state and status of Physical Education in South Africa (SA) furthermore highlighted that rural areas in SA have challenges in promoting motor skills development and subsequently active behavior because of limited resources, lack of qualified physical educators and poor sporting facilities [28].

However, the systematic review which includes 50 studies on the effectiveness of various interventions on childhood obesity [18], did not focus on the use of motor interventions to prevent obesity in children. A lack of information was therefore identified as a gap in the literature regarding the influence of early motor interventions to combat obesity in children, especially in rural areas. To this effect, the main aim of this study is to determine the immediate but also the sustainable influences of a 9-week movement program on the body composition of 7 to 8-year-old school children of the Eastern Cape Province.

## 2. Materials and Methods

### 2.1. Research Design

A quasi-experimental two group (control and intervention group), pre-post–re-test design was used in this study to evaluate the influence of a 9-week movement program on body composition and anthropometric characteristics of 7 to 8-year-old rural schoolchildren in the Raymond Mhlaba Municipality in the Eastern Cape of South Africa. After completion of the intervention, both groups were tested during post-intervention to determine the immediate effect on body composition. After six months of no intervention, both groups were re-tested again to determine the sustainability effects of the program on the body composition and anthropometric characteristics of the participants. The pre- and post-test as well as the intervention was conducted during the first part of the school year in the first school term (January to March 2022), while the re-test was carried out at the end of the third term before the school holidays commenced in September, 2022. The duration of the intervention program was considered against the background of the literature [29] which confirmed that the planned duration is long enough, (9 weeks, twice per week = 18 sessions), while the curriculum for *Physical Education in South Africa (Curriculum and Assessment Policy Statement* (CAPS) was also consulted as a guideline. However practical reasons also had to be taken into consideration as a school term in South Africa is only 12 weeks. This left us with 9 weeks for the intervention, as an additional three weeks for pre and post-testing were also needed, otherwise the holiday time would interfere with the results. As the intervention started at the beginning of the school year, we had to use the first week to recruit the participants and to wait for the consent forms to be returned in order to determine the number of participants that is ethically viable for participation in the study. Figure 1 shows randomization and participation in the study.

### 2.2. Ethical Aspects/Considerations

The following ethical issues were considered in this study: privacy, confidentiality, public trust, authorization and informed consent. The North-West University Health Research Ethical Committee (HREC) approved this study (NWU-00458-20-S1). Written informed consent was obtained from the Department of Basic Education of Raymond Mhlaba Municipality to conduct the study. The principals were asked permission to conduct the study at the school premises. A general meeting was held with the parents, to explain the aim of the study and to give them an opportunity to ask any questions. A standardized parental consent and child assent form had to be completed by parents and the children before participation was allowed. Children were informed that they were free to withdraw from the study at any stage without any penalty.

### 2.3. Study Population

The target population for this study was 7 to 8-year-old rural school children (grade 2) attending primary schools in Alice in the Raymond Mhlaba Municipality in the Eastern Cape of South Africa. Three primary schools were randomly selected from 7 schools to be involved in this study. A power analyses was performed by means of Statistica for Windows [30] to determine the number of participants that had to be recruited. One hundred and six school children were recruited, but only 93 provided consent, which represents a dropout rate of 12.2% at recruitment level. Random sampling again determined the participants that formed part of the intervention group and the control group. Sixty school children from the intervention group signed the consent forms. Fifty-seven school children (*n* = 27 boys; *n* = 30 girls) with mean ages of 6.96 (+0.68) at pre-test and 7.07 (+0.62) were tested during post-testing. At re-test, fifty-two participants (*n* = 24 boys; *n* = 28 girls) with a mean age of 7.72 (+0.67) were tested. The lost to follow-up dropout rate was 8.8% in this group. Reason for dropout was as a result of school children leaving the school. The control group included 46 school children from two schools who provided consent at pre-test (*n* = 17 boys; *n* = 19 girls) with a mean age of 7.36 (+0.68). At post-test, 35 children (*n* = 17 boys; *n* = 18 girls) with a mean age of 7.44 (+0.61) were assessed with a 2.8% dropout as a result of illness. Thirty children (*n* = 14 boys; *n* = 16 girls) were assessed during the re-testing with a mean age of 7.72 (+0.67) and with a 16.7% dropout. The dropout included absence from school on the days of testing and leaving the school.

### 2.4. Intervention Procedure

The 9-week fundamental movement skills program was presented during school hours to the intervention group after the pre-testing of both groups. The program included an 18-session movement intervention which was designed to improve the movement proficiency, body composition and fitness of young school children by focusing on improving basic motor skills and fitness. The SPARK (*Sports, Play & Active Recreation for Kids*) Physical Education program written by McKenzie and colleagues [31] was used as a guiding program to construct the movement program, but adapted with additional outcomes that addressed the intended outcomes of the specific program.

The program consisted of 5 min of warming up exercises, 30 min of core activities (locomotor, manipulation and fitness) and 5 min of cool down exercises. The sessions comprised of the 13 basic locomotor and manipulation motor skills (i.e., running, galloping, hopping, skipping, jumping, sliding, striking, bouncing, dribbling, catching, kicking, overhand throw and underhand throw). Aerobics exercises such as rope skipping, running, jumping jacks and traditional dance were additionally incorporated into the intervention program to improve body composition and fitness outcomes alongside the set movement proficiency outcomes. Children were placed into groups of five to encourage and ensure maximum participation and compliance of instructions. New groups of five were formed for each lesson to encourage socialization. The researcher and four well-trained research assistants led the FMS-based intervention program. The researcher also recorded the participants’ attendance in the intervention log. Translators were also provided to translate instructions to the participants who did not understand the English language.

After the study was completed, the control group participated in the lessons and the apparatus that was used during the intervention and the lesson plan were given to the control schools.

### 2.5. Instrumentation

#### 2.5.1. Anthropometry Measurements

Anthropometric data included stature, body mass, skinfolds (triceps and subscapular) and girth measurements (waist and hip girths), taken according to the standard procedures as prescribed by the International Society for the Advancement of Kinanthropometry (ISAK) [32]. Body mass was measured to the nearest 0.1 kg with an Omron (BF 511) electronic scale and body stature was measured to the nearest centimeter (cm) with a SECA stadiometer. The skinfolds were measured with a Harpenden skinfold caliper and the girth measurements with a Girths Lufkin non-extensible flexible steel anthropometric tape, both to the nearest 0.1 cm. The BMI of each of the participants was calculated using the two mean measurements weight kg/height m^2^. Children were classified in normal weight, overweight and obese categories using international age and gender adjusted BMI cut-off points provided by Cole and colleagues [33] for school-aged children and adolescents. All the measurements were taken by ISAK level 2 accredited anthropometrists in a private room.

#### 2.5.2. Anthropometric Derived Variables

The following indicators of body composition were derived from skinfold thickness, which is the sum of skinfold (triceps + subscapular). Different equations were used for boys and girls to calculate the body fat percentage. The formula for sum of skinfolds equal to or less than 35 mm was applied. This formula by Slaughter et al. [34] is internationally accepted for children of all ages.
BF% for boys = 1.21 (triceps + subscapular) − 0.008 (triceps + subscapular) ^2^ − 3.2
BF% for girls = 1.33 (triceps + subscapular) − 0.013 (triceps + subscapular) ^2^ − 2.5

### 2.6. Statistical Analyses

Statistical analysis was done using SPSS (version 27) [35]. A three-way-ANOVA type Hierarchical Linear Model (HLM) (mixed models) was used to test for intervention effects for which time (pre-, post- and retention), sex (boy and girl) and group (intervention and control) were used as predictors. The HLMs with child as subject were performed to take into account individual differences over time. The HLM is a better approach than repeated measures ANOVA since it can take missing data into account. No statistically significant difference was found in an ANOVA analysis for the school variable, therefore the school variable was not introduced as a separate level in the HLM. A statistical significance level of less than 5% (*p* < 0.05) was used for reporting significant effects, while levels at 10% (0.099) were also considered as marginal effects. An effect size was computed for each analysis using the Cohen’s-d to assess the practical significance of findings. The following Cohen’s cut points were used to interpret effect sizes: d > 0.2 = small, d > 0.5 = medium and d > 0.8 = large [36].

## 3. Results

Ninety-three (*N* = 93) children with a mean age of 7.12 (+0.71) years participated in the study. The intervention group comprised of 57 participants at pre-test and post-test and 52 participants for the re-test. The control group comprised of 36 participants at pre-test, 35 participants at post-test and 30 participants at re-test. The majority (98.9%) of the participants were black and 1.1% were of mixed ancestry (colored). The compliance to the intervention was high (97%) over the 18 lessons, conducted during PE periods, that comprised locomotor, ball control, balance and aerobics activities.

Table 1 provides the descriptive data of the anthropometric measurements stratified by group and time. This table shows that the intervention group had higher mean values of body mass (24.00 kg) compared to the control group (23.66 kg) at pre-test. A small significant reduction (*p* < 0.05) was noted in the body mass of the intervention group at post-testing. At post-test, the control group displayed a higher mean value (24.07 kg) in body mass than the intervention group (23.84 kg), which also significantly increased in the group, although both groups showed increases at re-test. Both groups displayed similar stature and changes in stature from the pre-testing to the re-testing.

There was a decrease in the mean values of triceps skinfold in both groups from pre-test to post-test. At re-testing, the control group had a higher mean values of triceps skinfold (8.93 mm) when compared to the intervention group (8.77 mm). The subscapular skinfold was higher in the control group from pre-test to re-test when compared to the intervention group. The mean value of waist circumference for the intervention group (53.34 cm) was higher when compared to the control group (52.21 cm) at pre-test. At post-test, the intervention group showed decreasing values (53.07 cm) but an increase (54.97 cm) in the re-test, while the control group showed an increasing trend from pre-test to re-test in their mean values (52.21 cm, 53.50 cm, 54.69 cm) of waist circumference respectively. Both groups displayed increasing mean values in the hip circumference from pre-test to re-test, although a decline was evident in the hip circumference of the intervention group (64.85 cm) at the post-test when compared to the control group (66.16 cm).

The intervention group had higher mean values in BMI at pre-test (16.17 kg/m^2^) when compared to the control group (15.79 kg/m^2^), although at post-test level they displayed a lower BMI level (15.57 kg/m^2^) compared to the control group (15.72 kg/m^2^). An increasing trend is also seen in the BMI of the control group from pre-test to re-test when compared to the intervention group. The fat percentage and the sum of skinfolds of both groups showed similar changes in both groups from pre-testing to re-testing, therefore intervention effects could not be seen in these measurements. With regards to significance of changes within groups, only three changes including body mass, waist circumferences and BMI were significant from pre- to post-testing and one from post-testing to re-testing (SSKF) and all significant changes were within the intervention group.

Table 2 provides the results of a Hierarchical Linear Model (HLM) analysis to determine intervention effects on all the anthropometric characteristics and the body composition for which time (pre-, post- and retention), sex (boy and girl) and group (intervention and control) were used as predictors. It also displays the means of the different anthropometric characteristics of boys and girls separately in the intervention and control groups over the three measuring points (Time 1 (T1), Time 2 (T2) and Time 3 (T3)). Table 3 reports the statistical and practical significance of mean changes of the boys and girls of both groups as displayed in Table 2, within the groups.

The three-way interaction analysis (group by time by sex) indicated that only the subscapular skinfold was significantly influenced by the intervention at different times of the intervention and that the intervention effects on boys and girls in the intervention group were also different to what was found in the control group (also see Table 3 that portrays the effect sizes of these changes, and a graphical representation of the changes in Figure 2a). In this regard during the post-testing, the subscapular skinfold of the boys in the control group increased, while it decreased in the girls of the same group. The two-way interaction analysis showed that only the subscapular skinfold was influenced by time and group, where this skinfold decreased in the intervention group (T1 to T3) while increasing in the control group over the same period.

A sustainable effect of the intervention is evident on this skinfold as both boys and girls in the intervention group had lower subscapular skinfolds at T3 compared to at T1, which was statistically significant for boys, also showing a small effect size for both boys and girls (*p* < 0.05; d = 0.1, *p* > 0.05; d = 0.2) (Table 3). The increase in the control group from (T1–T3) had an insignificant small effect sizes for both boys (*p* > 0.05; d = 0.0) and girls (*p* > 0.05; d = 0.1), respectively (Table 3) (Figure 2a). The waist circumference also showed intervention effects as a significant group by time (*p* = 0.054) and group by sex (*p* = 0.081) effects are seen (Table 2). Figure 2b clearly displays the decreasing waist circumference in the intervention group in both boys and girls at T2, compared to increasing tendencies in the control group. However, there was a significant small effect sizes on waist circumference from T1–T3 for both boys (*p* < 0.05; d = 0.4) and girls (*p* < 0.05; d = 0.3) in the intervention group (Table 3).

BMI was significantly influenced by the interaction of group by sex (*p* = 0.041). The BMI of the boys in the intervention group lowered from T1 to T2 (*p* < 0.05; d = 0.3) and to a point that was lower at T3 compared to at T1, while the BMI of boys in the control group increased from T1 to T2 (*p* > 0.05; d = 0.1) and from T1 to T3 (*p* > 0.05; d = 0.2). In the girls of the intervention group, BMI decreased at T2, which was significant with a small effect size (*p* < 0.01; d = 0.3), but increased at T3 with an insignificant small effect size (*p* > 0.05; d = 0.1) to almost a similar level than at T1. In the control group, the same changes were evident for girls from T1 to T2 with a marginal significant small effect size (*p* < 0.099; d = 0.1), although the mean value was higher at T3 compared to in T1, as is evident from Figure 2c. Group by time effects were also seen in BMI, where the intervention group showed decreasing and stabilizing effects on BMI compared to the control group, in which increasing BMI is seen over time (Table 2, Figure 2c).

In all the anthropometrics variables, the subscapular skinfold and BMI showed immediate and sustainable effects while the waist circumference showed an immediate effect in the intervention group relative to the control group.

Table 4 displays the changes in the BMI categories in the control and the intervention group over time (T1–T3). The intervention group, compared to the control group, displayed a higher overweight (14.04% vs. 2.78%) and obesity (5.26% vs. 2.78%) prevalence at pre-test. The prevalence of OW/OB was 19.30% in the intervention group compared to 5.56% in the control group. A decrease was noted in the number of overweight *n* = 8 to *n* = 5 (8.77%) and obese children *n* = 3 to *n* = 1 (1.75%) in the intervention group at post-testing, while the control group increased in overweight from *n* = 1 to *n* = 3 participants (8.57%) and 1 (2.86%) participant staying in the obese category at post-testing level. This resulted in a reduction of 8.78% in the prevalence of OW/OB in the intervention group (10.52%) when compared to a rise of 5.87% in the control group (11.43%) from pre- to post-testing after the intervention program. These results indicate a positive effect on the BMI of the participants in the intervention group, which is aligned with the statistical lowering of the BMI value in the group.

Regarding underweight, the results indicated that during pre-testing there was a slightly higher number of underweight *n* = 5 (8.77%) in the intervention group when compared to the control group *n* = 4 (11.11%). Also, 41 participants (71.93%) in the intervention group and 30 participants (83.33%) in the control group had normal weight.

After the intervention, during post-testing, a decrease was found in the number of underweight from *n* = 5 to *n* = 4 participants (7.02%) in the intervention group compared to an increase in the control group from *n* = 4 to *n* = 6 participants (17.14%). Likewise, there was an increase in the normal weight category in the intervention group, from *n* = 41 to *n* = 47 participants (82.5%), while a decrease was noted in the normal weight category in the control group from *n* = 30 to *n* = 25 participants (71.43%).

At re-testing, there was no record of underweight among participants in the intervention group, but one participant was recorded in the control group. The intervention group had 44 participants (84.62%), compared to 26 participants (86.67%) in the control group, who had normal weight. Furthermore, 13.46% (*n* = 7) participants were overweight and 1.92% (*n* = 1) participants obese in the intervention group compared to the control group, which had 10% (*n* = 3) participants overweight and non-obese. The prevalence of OW/OB in the intervention and control group after the retention testing were 15.38% and 10%, respectively.

## 4. Discussion

The purpose of this study was to investigate the influence of a 9-week movement program, consisting of 18 sessions on the body composition and anthropometric characteristics of 7–8-year-old school children. The outcome of the intervention showed positive benefits on body composition, skinfolds and circumferences, although not all the assessed skinfolds and circumferences improved. Significant improvements were found in BMI, the subscapular skinfold and waist circumference relative to the control group. These changes were furthermore sustained in BMI and the subscapular skinfold, although not in waist circumference. Interaction effects relating to sex were in addition found in the changes observed in these characteristics, which indicated that the body composition and anthropometric make-up of boys and girls, especially the subscapular skinfold, were differently influenced by the intervention.

Overall, the percentage of OW/OB among the participants (intervention and control group) in this study before the intervention signifies a high prevalence of OW and OB (13.98%) among school children, especially in rural areas, which is consistent with previous findings in South Africa [37,38]. Data from the most recent wave of the National Income Dynamics Study (NIDS 2017) shows that 16% of children aged 5–9 are classified as overweight or obese [10]. Prevalence of overweight or obesity in rural South African children between the ages 6–19 years in the Eastern Cape is furthermore reported at 16.9% [14] and 13.40% [39]. A study by Pienaar [16], in the North-West Province of South Africa, reported a prevalence increase of 4% in OW/OB from 12% to 16% in school children between the ages of 6 and 9 years. These statistics indicate different prevalence in different Provinces and high increases in prevalence in more recent studies in children from rural areas. In this regard, previous studies did predict a high prevalence of OW in rural areas, because of the consumption of sugar-sweetened beverages (SSBs), starchy and fatty foods and a more sedentary lifestyle and reduced daily PA [12,40]. The results of this study are positive, as it contributed to a lowering of prevalence of OW/OB in the intervention group relative to the control group. In our study, 19.30% of the intervention group and 5.56% of the control group were OW/OB at baseline. A reduction of 8.78% in the prevalence of OW/OB was, however, noted in the intervention group after participating in the program (19.30% to 10.52%), whereas an increase of 5.87% was observed in the prevalence of OW/OB in the control group (5.56% to 11.43%) (Table 4). This increase in overweight in the control group contributed to a higher overall OW/OB prevalence at post-testing in this group, in comparison to a significant decrease in the BMI level of the intervention group. A sustained lowering of the subscapular skinfold and a decrease in waist circumference in the intervention group additionally signified improved subcutaneous levels of fat in this group.

Circumferences are considered to be more reliable than skinfolds for estimating fat distribution and can always be measured regardless of body fatness and body size [41]. Compared to the reduction in the mean value of waist circumference and BMI in both boys and girls in the intervention group during post-testing, the control group had an increase in waist circumference in both boys and girls and an increase in BMI among boys which support an increase in unhealthy BMI. These improved body composition characteristics among the intervention group when compared to the control group was consistent with the intervention study by Zask and co-workers [42]. These researchers conducted the ‘Tooty Fruity Vegie’ program for 10-months among 560 children aged 3–6 years from the NSW North Coast area of Australia aiming to decrease overweight and obesity prevalence among children by improving FMS and decreasing unhealthy food consumption. A significant difference in waist circumference (−0.80 cm, *p* = 0.002) and a reduction in the BMI Z score (−0.15, *p* = 0.022) was found in the intervention group, compared to the control group.

Intervention effects were found on BMI, waist circumference and subscapular skinfold among boys and girls in the intervention group and both boys and girls in the intervention group also had sustainable effects in the subscapular skinfold compared to the control group. Boys in the intervention group had a sustainable effect on BMI which is again consistent with the findings of a previous study [43]. In that study [43], an intervention was conducted, consisting of 38 lessons (40–50 min), for 1 school year (March to November) during PE classes on 10.8-year-old participants from low socioeconomic areas of Melbourne, Australia. The intervention components based on behavioral modification (BM) and fundamental movement skills (FMS) have been previously been described and revealed a significant intervention effect on the BMI of the intervention group compared to the control group (−1.88 kg m^−2^, *p* < 0.01). These effects were also maintained after a 6 and 12 month-follow-up, but only among girls in the intervention group compared to the control group. Although this is contradictory to our study where boys in the intervention group had a sustainable effect, these findings confirm gender effects in intervention at young ages, therefore further research is needed where gender effects are taken into consideration, especially in young school children. However, the findings of another study [29], although conducted on younger children, revealed contrasting findings to our results. This study conducted an intense but short duration 4-week intervention (Monday through Friday) among 70 kindergartens and first grade school children of Northern Iowa, consisting of a 90 min after-school sessions that included motor skills. The immediate effects of this intervention revealed no significant change in waist circumference and BMI after a 4 months’ follow-up in both groups although, improved motor skills were found. Again, more research is recommended to address this inconsistent finding among studies.

The findings of the present study suggest that a short and organized movement intervention is sufficient to make a significant impact on certain characteristics of the body composition of school children. Changes in triceps skinfold cannot be attributed to the program, although it can be noted that changes in boys and girls were different over the three time points (pre-test to re-test). A longer duration of the program and the incorporating of these activity programs as part of PE lessons may be a recommendation to get more desirable results for all body composition characteristics of school children. The participants were happily involved in the aerobic activities that were part of the intervention such as rope skipping, kickbacks, running, jumping jacks and traditional dance, which might have contributed to their BMI reduction. These activities might have been appealing for the children in the intervention and might therefore have contributed to them using it in their everyday activities, again influencing the sustainable effect on BMI and the subscapular skinfold. Aerobic activities had the largest time allocation (7 min) while locomotor, ball control and balancing activities had 5 min each. It is suggested that the combination of FMS, balance and aerobic activities brought about the needed changes in BMI, as only fundamental movement activities might not have been sufficient to bring about these changes. The duration of the program can also be increased, with an increase in the time allocated for each activity, which can thereby increase the benefits of the program.

This study’s strengths were that it was the first to be conducted in rural schools in Raymond Mhlaba Municipality, South Africa, where the focus was on improving body composition through improving FMS in children growing up in rural areas. The findings provided valuable understanding of the challenges but also the benefits of implementing such programs in rural schools. A further strength is the 6-month follow-up period that examined the sustainable effects on the BMI and the anthropometric characteristics of school children. A study like this has not been carried out in this part of the Eastern Cape Province among school children from rural areas, therefore, it serves as one of its kind. There are also, limited studies that use FMS intervention to influence BMI of school children. Therefore, the intervention approach supported the theories indicating that competence in fundamental movement skills is related to various health benefits including increased PA [44,45,46] while it is also inversely associated with weight status [47]. On a psychological level, higher movement skill proficiency contributes to higher perceived competency that tracks through to adolescence and adulthood [48,49]. High perceived competence is reported to facilitate positive expectations for success, intrinsic motivation and achievement-oriented behaviors, such as engagement, effort to master skills, persistence in the face of difficulty and choice of challenging tasks [50]. These suggest an improved self-concept, mental health, activity levels, self-belief, social status among other peers, improved motivation and health-related quality of life, consequently contributing to the behavioral changes needed for the management of overweight and obesity, reducing excessive intake and promoting energy expenditure [48,49]. On a physiological level, engaging in a regular and sufficient levels of physical activity would improve body composition. Physical activities such as strengthening exercises would improve muscles mass and subsequently lean body mass, while aerobic activities would reduce adipose tissue, and therefore physical activity would lead to reduced body mass, adipose tissue and BMI [51]. This, therefore, is a promising strategy to prevent and improve overweight and obesity in school children. It is therefore suggested that programs of this nature may be necessary to be included into the PE curricula of primary schools to improve the currently non-functional PE curricula.

The study, however, also had limitations that need to be acknowledged. Only school children within the range of 7–8 years in Alice town participated in the study, therefore, care must be taken when generalizing the results to other areas or age groups. Further intervention studies are, therefore, recommended to be planned on similar aged school children, taking the limitations of this study into consideration, such as area, but also on older children. It should also be noted that the program did not address all the intended body composition and anthropometric components and showed different gender effects in some of the studied variables. Aspects that should be considered in this regard include the duration of the program, changing interest of children and the most suitable activities that can improve body composition. The program had to be fitted into the time allowed for PE at the intervention school, which was minimal and which affected the program. An increase in the awareness of schools of the importance of providing young children with adequate opportunities to be active through exposure to PE lessons is therefore important. In addition, the frequency of PE should be increased within the school curricula. Although aspects such as lack of facilities, equipment and resources and qualified PE teachers were not investigated, these could discourage teachers and create difficulty in implementing anything meaningful with regards to improving the health of young children through PE lessons. It is also recommended that the assessment of the children’s enjoyment of FMS intervention programs should be measured in similar studies, as such information can provide valuable information on the motivation of children and the content of the program to encourage participation. The sex differences that were noted in relation to body composition and anthropometric measurements during the study are, however, noteworthy at this young age and therefore further intervention studies are recommended to take this sex differences into consideration, especially among young children.

### Practical Implications

Given the persistent and high prevalence of overweight and obesity in school-aged children, it is important to consider all options to reduce childhood obesity. This study contributed to implementing an organized intervention program during PE for the intervention group and the outcome was positive, as there was a reduction in the prevalence of overweight and obesity. Intervention such as this is therefore considered to be essential to help prevent childhood obesity. However, although, the intervention was effective, only a small group of children (grade 2) benefited from the study. Large-scale government interventions are therefore required to enable primary strategies to prevent obesity and health adversity which are likely not to be inverted if there are no actions taken by the government. Government efforts towards promoting effective PE in rural school communities and providing facilities is important in improving children’s health and physical activities. Early prevention is, however, an important strategy that may prevent children from being exposed to obesogenic risky environments. In this study, evidence is provided that a preventive intervention with appropriate activities may help to control some risk such as increases in unhealthy BMI. Actions from individuals and school organizations are therefore necessary to help improve children’s quality of life.

## 5. Conclusions

The primary objective of this study was to implement an intervention that can effectively improve the BMI of school children within 9-weeks. A lowering prevalence of overweight and obesity (19.30% to 10.52%) was established among the participants in the intervention group, while increasing healthier BMI, skinfold thickness and circumferences also benefitted from the program. The results therefore show that a fundamental motor skills intervention combined with aerobic activities, designed to improve the body composition of school children at a young age is achievable with a low-cost intervention, by mainly incorporating these activities in the teaching curriculum of children. A movement intervention program can therefore promote healthy behavior and should be considered as a strategy to combat obesity in young children, especially in rural areas with limited resources.

## Figures and Tables

**Figure 1 ijerph-20-01762-f001:**
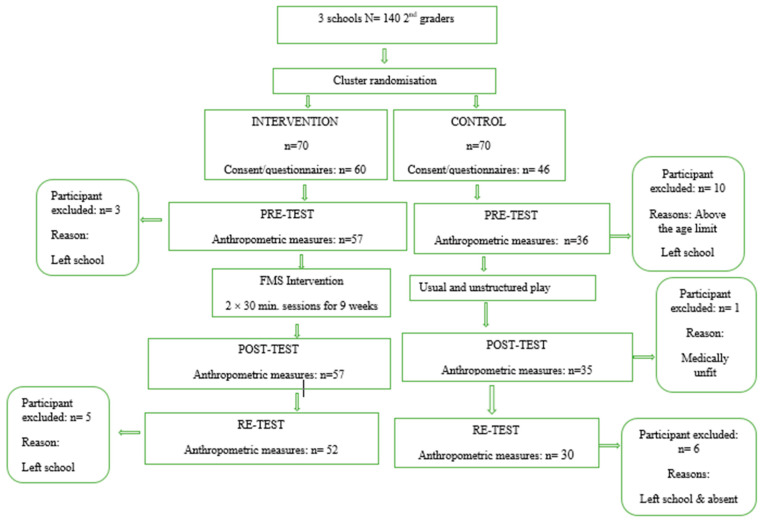
Flow diagram of recruitment, randomization and participation of school children in the study.

**Figure 2 ijerph-20-01762-f002:**
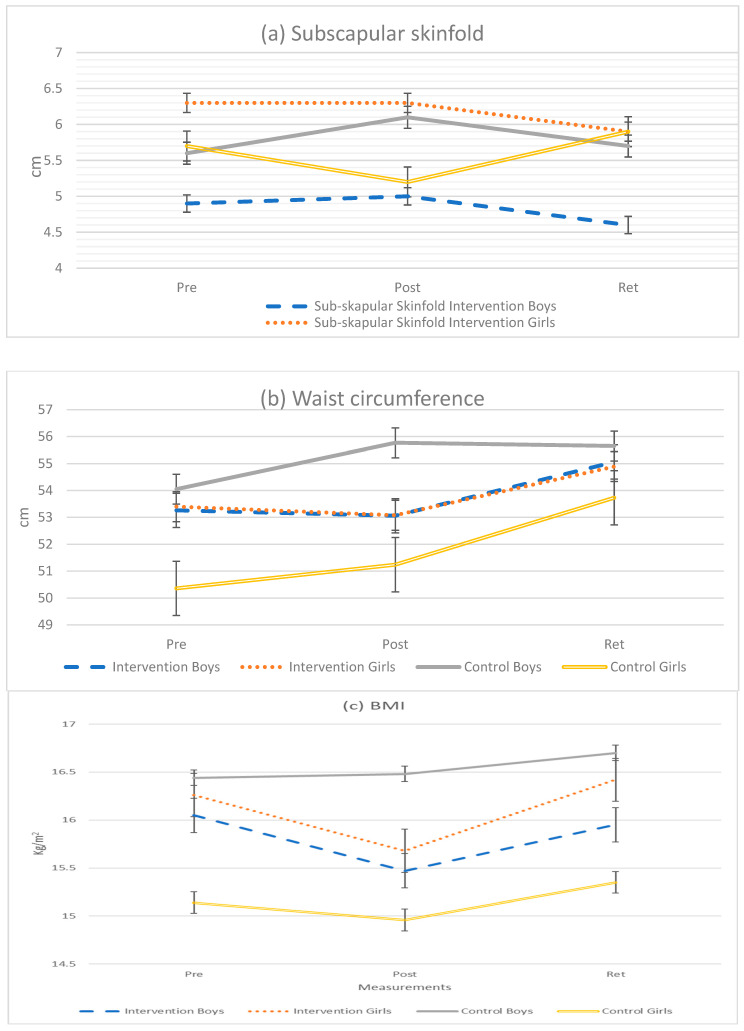
(**a**–**c**) Anthropometrics variables of participants according to sex, group and time.

**Table 1 ijerph-20-01762-t001:** Anthropometric characteristics and body composition of the participants (study sample) according to group and time.

Variables	Intervention Group						Control Group					
	Pre-Test		Post-Test		Re-Test		Pre-Test		Post-Test		Re-Test	
	Mean	Std Error	Mean	Std Error	Mean	Std Error	Mean	Std Error	Mean	Std Error	Mean	Std Error
Body mass (kg)	24.00	0.576	23.84 ^a–b,^*	0.576	25.63	0.578	23.66	0.725	24.07	0.726	25.37	0.730
Stature (cm)	121.81	0.741	123.41	0.741	125.43	0.742	122.08	0.933	123.45	0.934	125.48	0.935
TSKF (mm)	8.89	0.415	8.32	0.415	8.77	0.420	9.43	0.523	8.31	0.526	8.93	0.534
SSKF (mm)	5.59	0.366	5.65	0.366	5.27 ^b–c,^*	0.368	5.61	0.461	5.66	0.462	5.80	0.465
Waist circumference	53.34	0.635	53.07 ^a–b,^*	0.635	54.97	0.643	52.21	0.799	53.50	0.805	54.69	0.819
Hip circumference	63.07	0.712	64.85	0.712	66.32	0.717	61.60	0.895	66.16	0.899	66.64	0.908
BMI	16.17	0.266	15.57 ^a–b,^*	0.266	16.19	0.269	15.79	0.335	15.72	0.337	16.02	0.341
Fat %	12.82	0.696	12.49	0.696	11.04	0.696	13.58	0.876	12.15	0.886	10.36	0.871
Sum of SKF	14.48	0.796	13.98	0.796	12.78	0.796	15.04	1.002	13.53	1.011	11.84	0.998

TSKF = Triceps skinfold; SSKF = Subscapular skinfold; BMI = Body Mass Index; Fat % = Fat percentage, Sum of SKF = Sum of skinfold; ^a–b^ significant within group changes from pre-posttesting (*p* < 0.05); ^b–c^ = significant within group changes from post- to retention testing (*p* < 0.05). * = *p* < 0.05 (statistically significant).

**Table 2 ijerph-20-01762-t002:** Possible influences of co-variants on the anthropometric variables of the participants.

	Intervention Group						Control Group														
Variable	Boys Mean			Girls Mean			Boys Mean			Girls Mean			Variance		*p* Values						
	T1	T2	T3	T1	T2	T3	T1	T2	T3	T1	T2	T3	EST	Part	Group	Time	Group * Time	Sex	Group * Sex	Time * Sex	Group * Time * Sex
Body Mass	23.7	23.5	25.2	24.4	24.1	26.1	24.4	24.9	26.1	23.0	23.3	24.7	1.0	17.8	0.892	<0.01	0.141	0.687	0.222	0.712	0.943
Stature	121.7	123.1	125.2	121.9	123.7	125.7	121.4	122.5	124.6	122.8	124.4	126.4	0.6	30.6	0.919	<0.01	0.567	0.371	0.585	0.139	0.927
TSKF	7.3	7.1	7.2	10.5	9.6	10.4	9.1	8.0	8.6	9.7	8.7	9.3	1.6	8.2	0.718	<0.01	0.336	0.005	0.073	0.710	0.562
SSKF	4.9	5.0	4.6	6.3	6.3	5.9	5.6	6.1	5.7	5.7	5.2	5.9	0.5	7.1	0.748	0.587	0.051	0.341	0.201	0.026	0.045
Waist Cir	53.3	53.1	55.1	53.4	53.1	54.9	54.1	55.8	55.7	50.4	51.2	53.7	4.5	18.4	0.731	<0.01	0.054	0.079	0.081	0.211	0.112
Hip Cir.	61.4	63.7	64.8	64.7	66.0	67.8	61.9	66.6	67.0	61.3	65.7	66.3	3.1	25.7	0.961	<0.01	<0.01	0.317	0.108	0.425	0.880
BMI	16.1	15.5	16.0	16.3	15.7	16.4	16.4	16.5	16.7	15.1	15.0	15.4	0.5	3.5	0.747	<0.01	0.060	0.189	0.041	0.657	0.821
Fat %	10.3	10.2	8.4	15.3	14.8	13.6	12.5	12.1	9.0	14.7	12.2	11.7	15.4	12.2	0.921	<0.01	0.445	<0.01	0.076	0.376	0.696
Sum of SKF	12.2	12.1	10.5	16.7	15.9	15.1	14.7	14.2	11.1	15.4	12.9	12.6	15.1	21.0	0.802	<0.01	0.421	0.038	0.069	0.280	0.699

Pre-test; T2 = Post-test; Test 3 = Re-test; EST = Estimate of variance (MSE) s; Part = Participant; TSKF = Triceps skinfold; SSKF = Subscapular skinfold; Waist Cir. = Waist Circumference; Hip Cir. = Hip Circumference; BMI = Body Mass Index; Fat % = Fat percentage, Sum of SKF = Sum of skinfold, * = *p* < 0.05 (statistically significant).

**Table 3 ijerph-20-01762-t003:** Effect size of changes between measurements for all anthropometric and body-composition variables in both groups.

Intervention Group (Boys)					Intervention Group (Girls)			
Variables/Time	Mean Diff (SD)	df	*p*-Value	d	Mean Diff (SD)	df	*p*-Value	d
**Subscapular**								
**Pre-test—post-test**	−0.09 ± 0.84	26	0.571	0.0	−0.04 ± 0.92	29	0.828	0.0
**Post-test—re-test**	0.33 ± 0.77	23	0.046	0.2	0.38 ± 1.20	27	0.102	0.2
**Pre-test—re-test**	0.37 ± 0.74	23	0.023	0.1	0.31 ± 1.43	27	0.259	0.2
**Waist**								
**Pre-test—post-test**	0.22 ± 3.01	26	0.705	0.0	0.32 ± 2.64	29	0.516	0.1
**Post-test—re-test**	−1.97 ± 2.11	23	<0.01	0.4	−1.81 ± 3.85	27	0.019	0.4
**Pre-test—re-test**	−1.81 ± 3.14	23	0.010	0.4	−1.53 ± 3.51	27	0.029	0.3
**BMI**								
**Pre-test—post-test**	0.58 ± 1.46	26	0.048	0.3	0.61 ± 0.81	29	<0.01	0.3
**Post-test—re-test**	−0.52 ± 0.60	23	<0.01	0.3	−0.73 ± 1.17	27	0.003	0.4
**Pre-test—re-test**	0.13 ± 1.81	23	0.723	0.1	−0.15 ± 0.80	27	0.333	0.1
**Control group (Boys)**					**Control group (Girls)**			
**Subscapular**								
**Pre-test—post-test**	−0.54 ± 1.28	16	0.099	0.2	0.42 ± 0.94	17	0.076	0.2
**Post-test—re-test**	0.35 ± 1.24	13	0.312	0.2	−0.69 ± 0.83	14	0.006	0.3
**Pre-test—re-test**	−0.11 ± 0.91	13	0.666	0.0	−0.26 ± 0.72	15	0.173	0.1
**Waist**								
**Pre-test—post-test**	−1.60 ± 2.05	16	0.005	0.4	−0.96 ± 3.00	17	0.194	0.2
**Post-test—re-test**	−0.29 ± 2.77	13	0.699	0.0	−2.52 ± 3.75	14	0.021	0.5
**Pre-test—re-test**	−1.68 ± 2.13	13	0.011	0.3	−3.33 ± 2.90	15	<0.01	0.7
**BMI**								
**Pre-test—post-test**	−0.04 ± 1.02	16	0.871	0.1	0.18 ± 0.41	17	0.083	0.1
**Post-test—re-test**	−0.21 ± 0.40	13	0.067	0.1	−0.38 ± 0.67	14	0.044	0.2
**Pre-test—re-test**	−0.34 ± 1.05	13	0.243	0.2	−0.20 ± 0.62	15	0.217	0.2

df = degrees of freedom; d = effect size; d > 0.2 = small, d > 0.5 = medium and d > 0.8 = large.

**Table 4 ijerph-20-01762-t004:** Descriptive summary of the classification of the intervention group (IG) and control group (CG) in different BMI categories from pre-testing to re-testing.

BMI Classification	Total (93) Pre-Test		IG (57)		CG (36)		Total (92) Post-Test		IG (57)		CG (35)		Total (82) Re-Test		IG (52)		CG (30)	
	N	%	N	%	N	%	N	%	N	%	N	%	N	%	N	%	N	%
Underweight	9	9.68	5	8.77	4	11.11	10	10.87	4	7.02	6	17.14	1	1.22	0	0	1	3.33
Normal weight	71	76.34	41	71.93	30	83.33	72	78.26	47	82.5	25	71.43	70	85.37	44	84.62	26	86.67
Overweight	9	9.68	8	14.04	1	2.78	8	8.70	5	8.77	3	8.57	10	12.19	7	13.46	3	10
Obese	4	4.30	3	5.26	1	2.78	2	2.17	1	1.75	1	2.86	1	1.22	1	1.92	0	0
Overweight + obese	13	13.98	11	19.30	2	5.56	10	10.87	6	10.52	4	11.43	11	13.41	8	15.38	3	10

N = Total number; % = Percentage; IG = Intervention Group; CG = Control Group.

## Data Availability

The dataset is the property of the North-West University under the supervision of A.E. Pienaar. In this regard, A.E. Pienaar should be contacted if, for any reason, the data included in this paper needs to be shared. A.E. Pienaar is the principal investigator of this study.

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
