# Peer review of "The Influence of a 9-Week Movement Program on the Body Composition of 7- to 8-Year-Old Schoolchildren in the Eastern Cape of South Africa"

_ijerph, 2023, doi:10.3390/ijerph20031762_

Round 1
Reviewer 1 Report
1、in the abstract,the statement “anthropometric measurements and body mass” is not clear. Does anthropometric measurements include BMI?
2、In the introduction, the reason why you chose to use a 9-week intervention was not specified.
3、In terms of the statistical analysis, the reason why HLM was employed was not specified, and is “Cohen’s-d ” suitable as effect size?
4、In figure 1.,what do “1 school” and “2 school” represent?
5、In the discussion, the mechanism of the improvement has not been specified.
6、In "2.4 Intervention procedure" , the specific of method of changing group was not clear.
Reviewer 2 Report
Idamokoro et al. present a manuscript describing a study centred on a structured physical activity intervention in school-age children, with the objective of mediating unhealthy weight gain. Results of the study indicate that moderate beneficial changes in body composition were the outcome of the intervention, supporting the value of such physical activity programmes as a component in public health strategies to combat the obesity epidemic.
In its current form the manuscript contains only minor issues, which are as follows:
Page 2 line 71: 'negative health behaviors', It is not entirely clear what the authors mean by this.
Page 3 line 112: 'body image', Similarly, it is not entirely clear what the authors mean by this.
All tables, but particularly tables 2 and 3 could be improved for greater readability. For example, an alternating shading on rows would make reading across easier (e.g., table 2) and 'column' borders to distinguish data subsets (e.g., table 3). Additionally, please ensure that all relevant abbreviations are described in each table. For example, table 4, 'IG' and 'CG'.
Error bars are missing from panels A and B in fig. 2.
Page 16 line 452: FMS has previously been described.
Minor grammatic errors: line 457 '...research is needed...'; line 487 'Studies like this have...'; line 521 '... this one are...'
Round 2
Reviewer 1 Report
Some concerns have been solved. However,It would be better if you can add the physiological and psychological mechanism in the dicussion section.
